# Inferring Generative Model Structure
# with Static Analysis

**Paroma Varma**[1], **Bryan He**[2], **Payal Bajaj**[2],
**Nishith Khandwala**[2], **Imon Banerjee**[3], **Daniel Rubin**[3,4], **Christopher Ré**[2]
[1]Electrical Engineering, [2]Computer Science, [3]Biomedical Data Science, [4]Radiology
Stanford University
{paroma,bryanhe,pabajaj,nishith,imonb,rubin}@stanford.edu,
chrismre@cs.stanford.edu

## Abstract

Obtaining enough labeled data to robustly train complex discriminative models is a major bottleneck in the machine learning pipeline. A popular solution is combining multiple sources of weak supervision using generative models. The structure of these models affects the quality of the training labels, but is difficult to learn without any ground truth labels. We instead rely on weak supervision sources having some structure by virtue of being encoded programmatically. We present Coral, a paradigm that infers generative model structure by statically analyzing the code for these heuristics, thus significantly reducing the amount of data required to learn structure. We prove that Coral's sample complexity scales quasilinearly with the number of heuristics and number of relations identified, improving over the standard sample complexity, which is exponential in $n$ for learning $n^{\text{th}}$ degree relations. Empirically, Coral matches or outperforms traditional structure learning approaches by up to 3.81 F1 points. Using Coral to model dependencies instead of assuming independence results in better performance than a fully supervised model by 3.07 accuracy points when heuristics are used to label radiology data without ground truth labels.

## 1 Introduction

Complex discriminative models like deep neural networks rely on a large amount of labeled training data for their success. For many real-world applications, obtaining this magnitude of labeled data is one of the most expensive and time consuming aspects of the machine learning pipeline. Recently, generative models have been used to create training labels from various weak supervision sources, such as heuristics or knowledge bases, by modeling the true class label as a latent variable [1, 2, 27, 31, 36, 37]. After the necessary parameters for the generative models are learned using unlabeled data, the distribution over the true labels can be inferred. Properly specifying the structure of these generative models is essential in estimating the accuracy of the supervision sources. While traditional structure learning approaches have focused on the supervised case [23, 28, 41], previous works related to weak supervision assume that the structure is user-specified [1, 27, 31, 36]. Recently, Bach et al. [2] showed that it is possible to learn the structure of these models with a sample complexity that scales sublinearly with the number of possible binary dependencies. However, the sample complexity scales *exponentially* for higher degree dependencies, limiting its ability to learn complex dependency structures. Moreover, the time required to learn the dependencies also grows exponentially with the degree of dependencies, hindering the development of user-defined heuristics.

This poses a problem in many domains where high degree dependencies are common among heuristics that operate over a shared set of inputs. These inputs are interpretable characteristics extracted from the data. For example, various approaches in computer vision use predicted bounding box or segmentation

attributes [18, 19, 29], like location and size, to weakly supervise more complex image-based learning tasks [5, 7, 11, 26, 38]. Another example comes from the medical imaging domain, where attributes include characteristics such as the area, intensity and perimeter of a tumor, as shown in Figure 1. Note that these attributes are computationally represented, and the heuristics written over them are encoded programmatically as well. There are typically a relatively small set of interpretable characteristics, so the heuristics often share these attributes. This results in high order dependency structures among these sources, which are crucial to model in the generative model that learns accuracies for these sources.

To address the issue of learning higher order dependencies efficiently, we present Coral, a paradigm that statically analyzes the source code of the weak supervision sources to infer, rather than learn, the complex relations among heuristics. Coral's sample complexity scales quasilinearly with the number of relevant dependencies and does not scale with the degree of the dependency, unlike the sample complexity for Bach et al. [2], which scales exponentially with the degree of the dependency. Moreover, the time to identify these relations is constant in the degree of dependencies, since it only requires looking at the source code for each heuristic to find which heuristics share the same input. This allows Coral to infer high degree dependencies more efficiently than techniques that rely only on statistical methods to learn them, and thus generate a more accurate dependency structure for the heuristics. Coral then uses a generative model to learn the proper weights for this dependency structure to assign probabilistic labels to training data.

We experimentally validate the performance of Coral across various domains and show it outperforms traditional structure learning under various conditions while being significantly more computationally efficient. We show how modeling dependencies leads to an improvement of 3.81 F1 points compared to standard structure learning approaches. Additionally, we show that Coral can assign labels to data that have no ground truth labels, and this augmented training set results in improving the discriminative model performance by 3.07 points. For a complex relation-based image classification task, 6 heuristic functions written over *only* bounding box attributes as primitives are able to train a model that performs within 0.74 points of the F1 score achieved by a fully-supervised model trained on the rich, hand-labeled attribute and relation information in the Visual Genome database [21].

## 2   The Coral Paradigm

The Coral paradigm takes as input a set of domain-specific primitives and a set of programmatic user-defined heuristic functions that operate over the primitives. We formally define these abstractions in Section 2.1. Coral runs static analysis on the source code that defines the primitives and the heuristic functions to identify which sets of heuristics are related by virtue of sharing primitives (Section 2.2). Once Coral identifies these dependencies, it uses a factor graph to model the relationship between the heuristics, primitives and the true class label. We describe the conditions under which Coral can learn the structure of the generative model with significantly less data than traditional approaches in Section 2.3 and demonstrate how this affects generative model accuracy via simulations. Finally, we discuss how Coral learns the accuracies of the each heuristic and outputs probabilistic labels for the training data in Section 2.4.

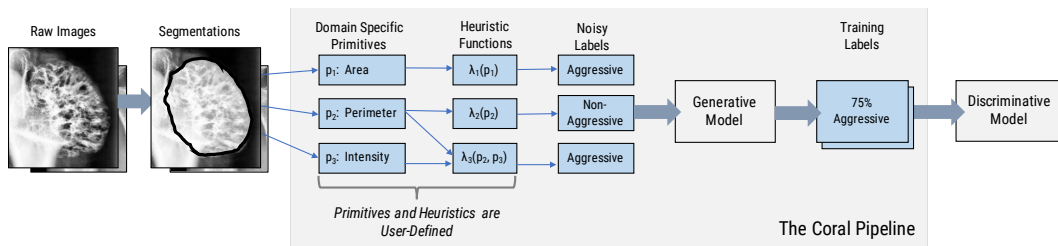

Figure 1: Running example for the Coral paradigm. Users apply standard algorithms to segment tumors from the X-ray and extract the *domain-specific primitives* from the image and segmentation. They write *heuristic functions* over the primitives that output a noisy label for each image. The generative model takes these as inputs and provides probabilistic training labels for the discriminative model.

## 2.1 Coral Abstractions

**Domain-Specific Primitives**    Domain-specific primitives (DSPs) in Coral are the simplest elements that heuristic functions take as input and operate over. DSPs in Coral have semantic meaning, making them interpretable for users. This is akin to the concept of language primitives in programming languages, in which they are the smallest unit of processing with meaning. The motivation for making the DSPs domain-specific instead of a general construct for the various data modalities is to allow users to take advantage of existing work in their field to extract meaningful characteristics from the raw data.

Figure 1 shows an example of a pipeline for bone tumor classification as aggressive or non-aggressive, inspired by one of our real experiments. First, an automated segmentation algorithm is used to generate a binary mask for where the tumor is [20, 25, 34, 39]. Then, we define 3 DSPs based on the segmentation: area ($p_1$), perimeter ($p_2$) and total intensity ($p_3$) of the segmented area. More complex characteristics such as those that capture texture, shape and edge features can also be used [4, 14, 22] (see Appendix).

We now define a formal construct for how DSPs are encoded programmatically. Users generate DSPs in Coral through a primitive specifier function, such as `create_primitives` in Figure 2(a). Specifically, this function takes as input a single unlabeled data point (and necessary intermediate representations such as the segmentation) and returns an instance of `PrimitiveSet`, which maps primitive names to primitive values, like integers (we refer to a specific instance of this class as P). Note that `P.ratio` is composed of two other primitives, while the rest of the primitives are generated independently from the image and segmentation.

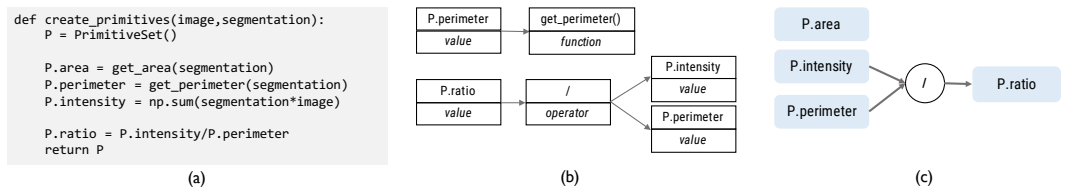

Figure 2: (a) The `create_primitives` function that generates primitives. (b) Part of the AST for the `create_primitives` function. (c) The composition structure that results from traversing the AST.

**Heuristic Functions**    In Coral, heuristic functions (HFs) can be viewed as mapping a subset of the DSPs to a noisy label for the training data, as shown in Figure 1. In our experience with user-defined HFs, we observe that HFs are usually nested if-then statements in which each statement checks whether the value of a single primitive or a combination of them are above or below a user-set threshold (see Appendix). As shown in Figure 3(a), they take fields of the object P as input and return a label (or abstain) based on the value of the input primitives. While our running example focuses on a single data point for DSP generation and HF construction, both procedures are applied to the *entire training set* to assign a set of noisy labels from each HF to each data point.

## 2.2 Static Dependency Analysis

Since the number of DSPs in some domains can be relatively small, multiple HFs can operate over the same DSPs. HFs that share at least one primitive are trivially related to each other. Prior work [2] learns these dependencies using the labels HFs assign to data points and its probability of success scales with the amount of data available. However, only *pairwise* HF dependencies can be learned efficiently, since the data required grows exponentially with the degree of the HF relation. This in turn limits the complexity of the dependency structure this method can accurately learn and model.

**Heuristic Function Inputs**    Coral takes advantage of the fact that users write HFs over a known, finite set of primitives. It *infers* dependencies that exist among HFs by simply looking at the source code of how the DSPs and HFs are constructed. This process *requires no data* to successfully learn the dependencies, making it more computationally efficient than standard approaches. In order to determine whether any set of HFs share at least one DSP, Coral looks at the input for each HF. Since the HFs only take as input the DSP they operate over, simply grouping HFs by the primitives they share is an efficient approach for recognizing these dependencies.

As shown in our running example, this would result in Coral not recognizing any dependencies among the HFs since the input for all 3 HFs are different (Figure 3(a)). This, however, would be incorrect, since the primitive `P.ratio` is composed of `P.perimeter` and `P.intensity`, which makes $\lambda_2$ and $\lambda_3$ related. Therefore, along with looking at the primitives that each HF takes as input, it is also essential to model how these primitives are *composed*.

**Primitive Compositions**    We use our running example in Figure 2 to explain how Coral gathers information about DSP compositions. Coral builds an abstract syntax tree (AST) to represent the computations the `create_primitives` function performs. An AST represents operations involving the primitives as a tree, as shown in Figure 2(b). To find primitive compositions from the AST, Coral first finds the expressions in the AST that add primitives to P (denoted in the AST as `P.name`). Then, for each assignment expression, Coral traverses the subtree rooted at the assignment expression and adds all other encountered primitives as a dependency for `P.name`. If no primitives are encountered in the subtree, the primitive is registered as being independent of the rest. The composition structure that results from traversing the AST is shown in Figure 2(c), where `P.area`, `P.intensity`, and `P.perimeter` are independent while `P.ratio` is a composition.

**Heuristic Function Dependency Structure**    With knowledge of how the DSPs are composed, we return to our original method of looking at the inputs of the HFs. As before, we identify that $\lambda_1$ and $\lambda_2$ use `P.area` and `P.perimeter`, respectively. However, we now know that $\lambda_3$ uses `P.ratio`, which is a composition of `P.intensity` and `P.perimeter`. This implies that $\lambda_3$ will be related to any HF that takes either `P.intensity`, `P.perimeter`, or both as inputs. We proceed to build a relational structure among the HFs and DSPs. As shown in Figure 3(b), this structure shows which *independent* DSPs each HF operates over. The relational structure implicitly encodes dependency information about the HFs — if an edge points from one primitive to $n$ HFs, those $n$ HFs are in an $n$-way relation by virtue of sharing that primitive. This dependency information can more formally be encoded in a factor graph shown in Figure 3(c), which is discussed in Section 2.3. Note that we chose a particular programmatic setup for creating DSPs and HFs to explain how static analysis can infer dependencies; however, this process can be modified to work with other setups that encode DSPs and HFs as well.

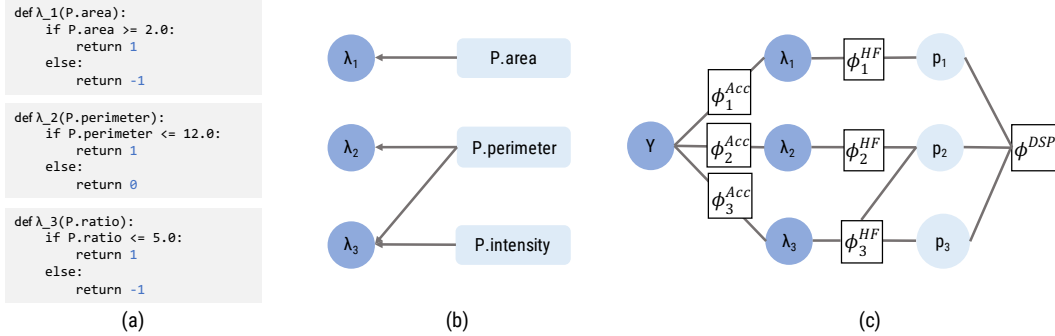

Figure 3: (a) shows the encoded HFs. (b) shows the HF dependency structure where DSP nodes have an edge going to the HFs that use them as inputs (explicitly or implicitly). (c) shows the factor graph Coral uses to model the relationship between HFs, DSPs, and latent class label Y.

## 2.3    Creating the Generative Model

We now describe the generative model used to predict the true class labels. The Coral model uses a factor graph (Figure 3(c)) to model the relationship between the primitives ($p \in \mathbb{R}$), heuristic functions ($\lambda \in \{-1,0,1\}$) and latent class label ($Y \in \{-1,1\}$). We show that by incorporating information about how primitives are shared across HFs from static analysis, this factor graph infers all dependencies between the heuristics that are guaranteed to be present. We also describe how Coral recovers additional dependencies among the heuristics by studying empirical relationships between the primitives.

**Modeling Heuristic Function Dependencies**    Now that dependencies have been inferred via static analysis, the goal is to learn the accuracies for each HF and assign labels to training data accordingly.

The factor graph thus consists of two types of factors: accuracy factors $\phi^{\text{Acc}}$ and HF factors from static analysis $\phi^{\text{HF}}$.

The accuracy factors specify the accuracy of each heuristic function and are defined as
$$\phi_i^{\text{Acc}}(Y,\lambda_i) = Y\lambda_i,\, i = 1,...,n$$
where $n$ is the total number of heuristic functions.

The static analysis factors ensure that the heuristics are correctly evaluated based on the HF dependencies found via static analysis. They ensure that a probability of zero is given to any configuration where an HF does not have the correct value given the primitives it depends on. The static analysis factors are defined as.
$$\phi_i^{\text{HF}}(\lambda_i, p_1,...,p_m) = \begin{cases} 0 & \text{if } \lambda_i \text{ is valid given } p_1,...,p_m \\ -\infty & \text{otherwise} \end{cases},\, i = 1,...,n$$
Since these factors are obtained directly from static analysis, *they can be recovered with no data*.

However, we note that static analysis is not sufficient to capture all dependencies required in the factor graph to accurately model the process of generating training labels. Specifically, static analysis can

  (i)  pick up spurious dependencies among HFs that are not truly dependent on each other, or
  (ii) miss key dependencies among HFs that exist due to dependencies among the DSPs in the HFs.

(i) can occur if some $\lambda_A$ takes as input DSPs $p_i, p_j$ and $\lambda_B$ takes as input DSPs $p_i, p_k$, but $p_i$ always has the same value. Although static analysis would pick up that $\lambda_A$ and $\lambda_B$ share a primitive and should have a dependency, this may not be true if $p_j$ and $p_k$ are independent. (ii) can occur if two HFs depend on different primitives, but these primitives happen to always have the same value. In this case, it is impossible for static analysis to infer the dependency between the HFs if the primitives have different names and are generated independently, as described in Section 2.2. A more realistic scenario comes from our running example, where we would expect the area and perimeter of the tumor to be related.

To account for both cases, it is necessary to capture the possible dependencies that occur among the DSPs to ensure that the dependencies from static analysis do not misspecify the factor graph. We introduce a factor to account for additional dependencies among the primitives, $\phi^{\text{DSP}}$. There are many possible choices for this dependency factor, but one simple choice is to model pairwise similarity between the primitives. For binary and discrete primitives, the dependency factor with pairwise similarity can be represented as
$$\phi^{\text{DSP}}(p_1,...,p_m) = \sum_{i<j} \phi_{ij}^{\text{Sim}}(p_i, p_j), \text{ where } \phi_{ij}^{\text{Sim}}(p_i, p_j) = \mathbb{I}[p_i = p_j].$$

The dependency factor can be generalized to continuous-valued primitives by binning the primitives into discrete values before comparing for similarity.

Finally, with three types of factors, the probability distribution specified by the factor graph is
$$P(y, \lambda_1,...,\lambda_n, p_1,...,p_m) \propto \exp\left( \sum_{i=1}^{n} \theta_i^{\text{Acc}}\phi_i^{\text{Acc}} + \sum_{i=1}^{n} \phi_i^{\text{HF}} + \sum_{i=1}^{m}\sum_{j=i+1}^{m} \theta_{ij}^{\text{Sim}}\phi_{ij}^{\text{Sim}} \right)$$
where $\theta^{\text{Acc}}$ and $\theta_{ij}^{\text{Sim}}$ are weights that specify the strength of factors $\phi^{\text{Acc}}$ and $\phi_{ij}^{\text{Sim}}$.

**Inferring Dependencies without Data** The HF factors capture all dependencies among the heuristic functions that are not represented by the $\phi^{\text{DSP}}$ factor. The dependencies represented by the $\phi^{\text{DSP}}$ factor are precisely the dependencies that cannot be inferred via static analysis due to the fact that this factor depends solely on the content of the primitives. It is therefore impossible to determine what this factor is without data.

While assuming that we have the true $\phi^{\text{DSP}}$ seems like a strong condition, we find that in real-world experiments, including the $\phi^{\text{DSP}}$ factor rarely leads to improvements over the case when we only include the $\phi^{\text{Acc}}$ and $\phi^{\text{HF}}$ factors. In some of our experiments (see Section 3), we use bounding box location, size and object labels as domain-specific primitives for image and video querying tasks. Since these primitives are not correlated, modeling the primitive dependency does not lead to any improvement over just modeling HF dependencies from static analysis. Moreover, in other experiments where modeling the relation among primitives helps, we observe relatively small benefits above what modeling HF dependencies provides (Section 3). Therefore, even without data, it is possible to model the most important dependencies among HFs that lead to significant gains over the case in which no dependencies are modeled.

## 2.4 Generating Probabilistic Training Labels

Given the probability distribution of the factor graph, our goal is to learn the proper weights $\theta_i^{\text{Acc}}$ and $\theta_{ij}^{\text{Sim}}$. Coral adopts structure learning approaches described in recent work [2], which learns dependency structures in the weak supervision setting and maximizes the $\ell_1$-regularized marginal pseudolikelihood of each primitive to learn the weights of the relevant factors.

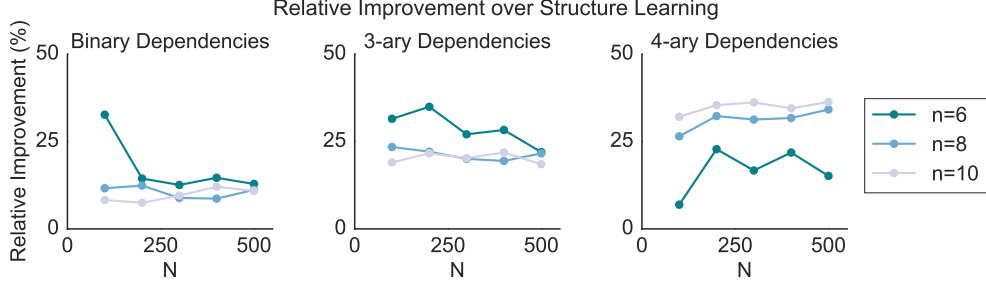

Figure 4: Simulation demonstrating improved generative model accuracy with Coral compared to structure learning [2] and Coral. Relative improvement of Coral over structure learning is plotted against number of unlabeled data points ($N$) and number of HFs ($n$).

To learn the weights of the generative model, we use contrastive divergence [15] as a maximum likelihood estimation routine and maximize the marginal likelihood of the observed primitives. Gibbs sampling is used to estimate the intractable gradients, which are then used in stochastic gradient descent. Because the HFs are typically deterministic functions of the primitives (represented as the $-\infty$ value of the correctness factors for invalid heuristic values), standard Gibbs sampling will not be able to mix properly. As a result, we modify the Gibbs sampler to simultaneously sample one primitive along with all heuristics that depend on it. Despite the fact that the true class label is latent, this process still converges to the correct parameter values [27]. Additionally, the amount of data necessary to learn the parameters scales quasilinearly with the number of parameters. In our case, the number of parameters is simply the number of heuristics $n$ and the number of relevant primitive similarity dependencies $s$.

We now formally state the conditions for this result, which match those of Ratner et al. [27], and give the sample complexity of our method. First, we assume that there exists some feasible parameter set $\Theta \subset \mathbb{R}^n$ that is known to contain the parameter $\theta^* = (\theta^{\text{Acc}}, \theta^{\text{Sim}})$ that models the true distribution $\pi^*$ of the data:

$$\exists \theta^* \in \Theta \text{ s.t. } \forall \pi^*(p_1,...,p_m,Y) = \mu_\theta(p_1,...,p_m,Y). \tag{1}$$

Next, we must be able to accurately learn $\theta^*$ if we are provided with labeled samples of the true distribution. Specifically, there must be an asymptotically unbiased estimator $\hat{\theta}$ that takes some set of labeled data $T$ independently sampled from $\pi^*$ such that for some $c > 0$,

$$\text{Cov}\left(\hat{\theta}(T)\right) \preceq (2c|T|)^{-1} I. \tag{2}$$

Finally, we must have enough sufficiently accurate heuristics so that we have a reasonable estimate of Y. For any two feasible models $\theta_1, \theta_2 \in \Theta$,

$$\mathbb{E}_{(p_1,...,p_m,Y) \sim \mu_{\theta_1}}\left[\text{Var}_{(p_1',...,p_m',Y') \sim \mu_{\theta_2}}(Y'|p_1=p_1',...,p_m=p_m')\right] \leq \frac{c}{n+s} \tag{3}$$

**Proposition 1.** *Suppose that we run stochastic gradient descent to produce estimates of the weights $\hat{\theta} = (\hat{\theta}^{Acc}, \hat{\theta}^{Sim})$ in a setup satisfying conditions (1), (2), and (3). Then, for any fixed error $\epsilon > 0$, if the number of unlabeled data points $N$ is at least $\Omega[(n+s)\log(n+s)]$, then our expected parameter error is bounded by $\mathbb{E}\left[\|\hat{\theta} - \theta^*\|^2\right] \leq \epsilon^2$.*

The proof follows from the sample complexity of Ratner et al. [27] and appears in the Appendix. With the weights $\hat{\theta}_i^{\text{Acc}}$ and $\hat{\theta}_{ij}^{\text{Sim}}$ maximizing the marginal likelihood of the observed primitives, we have a fully specified factor graph and complete generative model, which can be used to predict the latent class label. For each data point, we compute the label each heuristic function applies to it using the

values of the domain-specific primitives. Through the accuracy factors, we then estimate a distribution for the latent class label and use these noisy labels to train a discriminative model.

We present a simulation to empirically compare our sample complexity with that of structure learning [2]. In our simulation, we have $n$ HFs, each with an accuracy of $75\%$, and explore settings in which there exists one binary, 3-ary and 4-ary dependency among the HFs. The dependent HFs share exactly one primitive, and the primitives themselves are independent ($s=0$). We show our results in Figure 4. In the case with a binary dependency, structure learning recovers the necessary dependency with few samples, and has similar performance to Coral. In contrast, in the second and third settings with high-order dependencies, structure learning struggles to recover the relevant dependency, and performs worse than Coral even as more training data is provided.

## 3   Experimental Results

We seek to experimentally validate the following claims about our approach. Our first claim is that HF dependencies inferred via static analysis perform significantly better than a model that does not take dependencies into account. Second, we compare to a structure learning approach for weak supervision [2] and show how we outperform it over a variety of domains. Finally, we show that in case primitive dependencies exist, Coral can learn and model those as well. We show that modeling the dependencies between the heuristic functions and primitives can generate training sets that, in some cases, beat fully supervised models by labeling additional unlabeled data. Our classification tasks range from specialized medical domains to natural images and video, and we include details of the DSPs and HFs in the Appendix. Note that while the number of HFs and DSPs is fairly low (Table 1), using static analysis to automatically infer dependencies rather than ask users to identify them saves significant effort since the number of possible dependencies grows exponentially with the number of HFs present.

We compare our approach to majority vote (MV), generative models that learn the accuracies of different heuristics, specifically one that assumes the heuristics are independent (Indep) [27], and Bach et al. [2] that *learns* the binary inter-heuristic dependencies (Learn Dep). We also compare to the fully supervised (FS) case, and measure the performance of the discriminative model trained with labels generated using the above methods. We split our approach into two parts: inferring HF dependencies using only static analysis (HF Dep) and additionally learning primitive level dependencies (HF+DSP Dep).

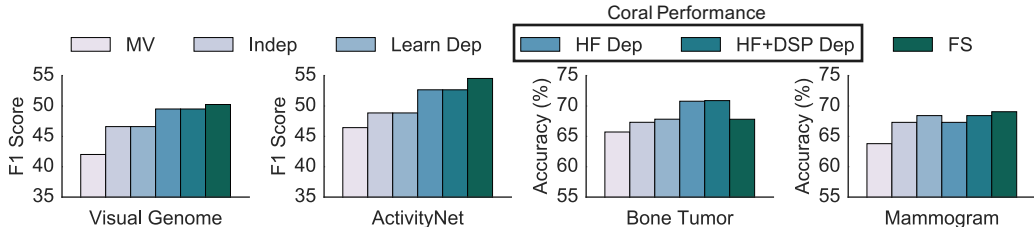

Figure 5: Discriminative model performance comparing HF Dep (HF dependencies from static analysis) and HF+DSP Dep (HF and DSP dependencies) to other methods. Numbers in Appendix.

**Visual Genome and ActivityNet Classification**   We explore how to extract complex relations in images and videos given object labels and their bounding boxes. We used subsets of two datasets, Visual Genome [21] and ActivityNet [9], and defined our task as finding images of "a person biking down a road" and finding basketball videos, respectively. For both tasks, a small set of DSPs were shared heavily among HFs, and modeling the dependencies observed by static analysis led to a significant improvement over the independent case. Since these dependencies involved groups of 3 or more heuristics, Coral improved significantly over structure learning as well, which was unable to model these dependencies due to the lack of enough data. Moreover, modeling primitive dependencies did not help since the primitives were indeed independent (Table 1). We report our results for these tasks in terms of the F1 score (harmonic mean of the precision and recall) since there was significant class imbalance which accuracy would not capture well.

**Bone Tumor Classification**   We used a set of 802 labeled bone tumor X-ray images along with their radiologist-drawn segmentations. Our task was to differentiate between aggressive and non-aggressive

Table 1: Heuristic Function (HF) and Domain-Specific Primitive (DSP) statistics. Discriminative model improvement with HF+DSP Dep over other methods. *improvements shown in terms of F1 score, rest in terms of accuracy. ActivityNet model is LR using VGGNet embeddings as features.

| Application | Number of | | | Model | Improvement Over | | | |
|---|---|---|---|---|---|---|---|---|
| | DSPs | HFs | Shared DSPs | | MV | Indep | Learn Dep | FS |
| Visual Genome | 7 | 5 | 2 | GoogLeNet | 7.49* | 2.90* | 2.90* | -0.74* |
| ActivityNet | 5 | 4 | 2 | VGGNet+LR | 6.23* | 3.81* | 3.81* | -1.87* |
| Bone Tumor | 17 | 7 | 0 | LR | 5.17 | 3.57 | 3.06 | 3.07 |
| Mammogram | 6 | 6 | 0 | GoogLeNet | 4.62 | 1.11 | 0 | -0.64 |

tumors. We generated HFs that were a combination of hand-tuned rules and decision-tree generated rules (tuned on a small held out subset of the dataset). The discriminative model utilized a set of 400 hand-tuned features (note that there is no overlap between these features and the DSPs) that encoded various shape, texture, edge and intensity-based characteristics. Although there were no explicitly shared primitives in this dataset, the generative model was still able to model the training labels more accurately with knowledge of how heuristics used primitives, which affects the relative false positive and false negative rates. Thus, the generative model significantly improved over the independent model. Moreover, a small dataset size hindered structure learning, which gave a minimal boost over the independent case (Table 1). When we used heuristics in Coral to label an additional 800 images that had no ground truth labels, we beat the previous FS score by 3.07 points (Figure 5, Table 1).

**Mammogram Tumor Classification** We used the DDSM-CBIS [32] dataset, which consists of 1800 scanned film mammograms and associated segmentations for the tumors in the form of binary masks. Our task was to identify whether a tumor is malignant or benign, and each heuristic only operated over one primitive, resulting in no dependencies that static analysis could identify. In this case, structure learning performed better than Coral when we only used static analysis to infer dependencies (Figure 5). However, including primitive dependencies allowed us to match structure learning, resulting in a 1.11 point improvement over the independent case (Figure 5, Table 1).

## 4   Related Work

As the need for labeled training data grows, a common alternative is to utilize weak supervision sources such as distant supervision [10, 24], multi-instance learning [16, 30], and heuristics [8, 35]. Specifically for images, weak supervision using object detection and segmentation or visual databases is a popular technique as well (detailed discussion in Appendix). Estimating the accuracies of these sources without access to ground truth labels is a classic problem [13]. Methods such as crowdsourcing [12, 17, 40], boosting[3, 33], co-training [6], and learning from noisy labels are some of the popular approaches that can combine various sources of weak supervision to assign noisy labels to data. However, Coral does not require *any* labeled data to model the dependencies among the heuristics, which can be interpreted as workers, classifiers or views for the above methods, and domain-specific primitives.

Recently, generative models have also been used to combine various sources of weak supervision [1, 31, 36, 37]. One specific example, data programming [27], proposes using multiple sources of weak supervision for text data in order to describe a generative model and subsequently learns the accuracies of these sources. Coral also focuses on multiple programmatically encoded heuristics that can weakly label data and learns their accuracies to assign labels to training data. However, Coral adds an additional layer of domain-specific primitives in its generative model, which allows it to generalize beyond text-based heuristics. It also *infers* the dependencies among the heuristics and the primitives, rather than requiring users to specify them.

Other previous work also assume that this structure in generative models is user-specified [1, 31, 36]. However, Bach et al. [2] recently showed that it is possible to learn the dependency structure among sources of weak supervision with a sample complexity that scales sublinearly with the number of possible pairwise dependencies. Coral instead identifies the dependencies among the heuristic functions by inspecting the content of the programmable functions, therefore relying on significantly less data to learn the generative model structure. Moreover, Coral can also pick up higher-order dependencies, for which Bach et al. [2] needs large amounts of data to detect.

# 5 Conclusion and Future Work

In this paper, we introduced Coral, a paradigm that models the dependency structure of weak supervision heuristics and systematically combines their outputs to assign probabilistic labels to training data. We described how Coral takes advantage of the programmatic nature of these heuristics in order to infer dependencies among them via static analysis. Coral therefore requires a sample complexity that is quasilinear in the number of heuristics and relations found. We showed how Coral leads to significant improvements in discriminative model accuracy over traditional structure learning approaches across various domains. Coral scratches the surface of the possible ways weak supervision can borrow from the field of programming languages, especially as weak supervision sources are used to label large magnitudes of data and need to be encoded programmatically. We look at a natural extension of treating the process of encoding heuristics as writing functions and hope to explore the interactions between systematic training set creation and concepts from the programming language field.

**Acknowledgments** We thank Shoumik Palkar, Stephen Bach, and Sen Wu for their helpful conversations and feedback. We are grateful to Darvin Yi for his assistance with the DDSM dataset based experiments and associated deep learning models. We acknowledge the use of the bone tumor dataset annotated by Drs. Christopher Beaulieu and Bao Do and carefully collected over his career by the late Henry H. Jones, M.D. (aka "Bones Jones"). This material is based on research sponsored by Defense Advanced Research Projects Agency (DARPA) under agreement number FA8750-17-2-0095. We gratefully acknowledge the support of the DARPA SIMPLEX program under No. N66001-15-C-4043, DARPA FA8750-12-2-0335 and FA8750-13-2-0039, DOE 108845, the National Science Foundation (NSF) Graduate Research Fellowship under No. DGE-114747, Joseph W. and Hon Mai Goodman Stanford Graduate Fellowship, National Institute of Health (NIH) U54EB020405, the Office of Naval Research (ONR) under awards No. N000141210041 and No. N000141310129, the Moore Foundation, the Okawa Research Grant, American Family Insurance, Accenture, Toshiba, and Intel. This research was supported in part by affiliate members and other supporters of the Stanford DAWN project: Intel, Microsoft, Teradata, and VMware. The U.S. Government is authorized to reproduce and distribute reprints for Governmental purposes notwithstanding any copyright notation thereon. The views and conclusions contained herein are those of the authors and should not be interpreted as necessarily representing the official policies or endorsements, either expressed or implied, of DARPA or the U.S. Government. Any opinions, findings, and conclusions or recommendations expressed in this material are those of the authors and do not necessarily reflect the views of DARPA, AFRL, NSF, NIH, ONR, or the U.S. government.

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
