[Supplementary Material · coral_nips_supp.pdf]

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

# A Extended Related Work

A popular approach for creating training sets is to provide weak or distant supervision to label data based on information from a knowledge base [12, 30], crowdsourcing [15], heuristic patterns [10, 19], user input [42], a set of user-defined labeling functions [34], or hand-engineered constraints [43]. Our inspiration for Coral came from observing various weak supervision and image description techniques developed in the field of computer vision. Early work looked at describing images in terms of a set of primitives to find instances of described objects in images [9, 16]. More recently, learned characteristics such as bounding boxes from object detection and areas from image segmentation have been used in order to weakly supervise more complex image-based learning tasks [6, 8, 13, 32, 46]. Moreover, the recent development of a 'knowledge base' for images, Visual Genome [26], has provided access to an image database with rich, crowdsourced attribute and relational information. This data in turn powers other methods that rely on the information in Visual Genome to supervise other tasks [17, 28]. This trend of using the possibly noisy information about the data at hand in order to weakly supervise models that look for complex relationships is one of the main motivations for Coral, which is a domain-agnostic method that can combine various sources of weak supervision while modeling the relationship among the sources as well as the primitives they operate over.

A similar methodology of using interpretable characteristics of images for classification tasks exists in the medical field as well [5, 33]. Other techniques such as content-based image retrieval (CBIR) rely on low-level, quantitative features of images, such as tumor texture, obtained from image analysis techniques to query similar images in a database of radiology images. However, there remains a gap between these content-based primitives and semantic, interpretable primitives that humans use in order to describe these images [27]. Coral could combine the two sources of supervision in order to build a system that combines the two methodologies in the optimal manner, while also helping discover correlations among the semantic and quantitative primitives.

# B Additional Experimental Results

We present the results from Section 3 in a table format here. We also provide a list of the domain-specific primitives and heuristic functions used in each of our experiments, with a short description of what they mean.

Table 2: Numbers for Figure 5(*ed have F1 scores, rest are accuracy). MV: majority vote; Indep: assume heuristics independent; Learn: model learned dependencies; HF-Dep: model heuristic dependencies identified using static analysis; HF+DSP Dep: additionally model learned primitive dependencies; FS: fully supervised

| Application | MV | Indep | Learn Dep | HF Dep | HF+DSP Dep | FS |
|---|---|---|---|---|---|---|
| Visual Genome* | 42.02 | 46.61 | 46.61 | 49.51 | 49.51 | 50.25 |
| ActivityNet* | 46.44 | 48.86 | 48.86 | 52.67 | 52.67 | 54.54 |
| Bone Tumor | 65.72 | 67.32 | 67.83 | 70.79 | 70.89 | 67.82 |
| Mammogram | 63.8 | 67.31 | 68.42 | 67.31 | 68.42 | 69.06 |

# C  Proof of Proposition 1

**Proposition 1.** *Suppose that we run stochastic gradient descent to produce estimates of the weights $\hat{\theta} = (\hat{\theta}^{Acc}, \hat{\theta}^{Sim})$ in a setup satisfying conditions (1), (2), and (3). Then, for any fixed error $\epsilon > 0$, if the number of unlabeled data points $N$ is at least $\Omega[(n+s)\log(n+s)]$, then our expected parameter error is bounded by $\mathbb{E}\left[\|\hat{\theta} - \theta^*\|^2\right] \leq \epsilon^2$.*

*Proof.* The proof of proposition 1 closely follows Theorem 2 of Ratner et al. [34]. First, notice that all of the necessary conditions of this theorem are satisfied. The number of weights to be learned in this theorem is $M$. In our setting, $M = n+s$. Notice that if we have at least $\Omega[(n+s)\log(n+s)]$ unlabeled data points, then we satisfy the conditions of the theorem. As a result, the bound on the expected parameter error directly follows. $\square$

Table 3: Domain-specific Primitives for Mammogram Tumor Classification

| Primitives | Type | Description |
| --- | --- | --- |
| area | float | area of the tumor |
| diameter | float | diameter of the tumor |
| eccentricity | float | eccentricity of the image |
| perimeter | float | perimeter of the tumor |
| max_intensity | float | maximum intensity of the tumor |
| mean_intensity | float | mean intensity of the tumor |

Table 4: Domain-specific Primitives for Image Querying

| Primitives | Type | Description |
| --- | --- | --- |
| person | bool | image contains a person/man/woman |
| person.position | (float,float) | coordinates of the person |
| person.area | float | area of bounsing box of person |
| road | bool | image contains a road/street |
| car | bool | image contains car/bus/truck |
| bike | bool | image contains a bike/bicycle/cycle |
| bike.position | (float,float) | coordinates of the bike |
| bike.area | float | area of bounding box of bike |

Table 5: Domain-specific Primitives for Video Classification

| Primitives | Type | Description |
| --- | --- | --- |
| person | bool | image contains a person |
| ball | bool | image contains sports ball |
| (person.top,person.bottom) | (float,float) | top and bottom coordinates of the person |
| (ball.top,ball.bottom) | (float,float) | top and bottom coordinates of the ball |
| ball.color | (float, float, float) | R,G,B colors of the ball |
| vertical_distance | float | cumulative absolute difference in ball.top values over frames |

Table 6: Domain-specific Primitives for Bone Tumor Classification

| Primitives | Type | Description |
|---|---|---|
| daube_hist_164 | float | Daubechies features |
| daube_hist_224 | float | Daubechies features |
| daube_hist_201 | float | Daubechies features |
| window_std | float | |
| window_median | float | |
| scale_median | float | |
| lesion_density | float | quantify edge sharpness along the lesion contour |
| edge_sharpness | float | quantify edge sharpness along the lesion contour |
| equiv_diameter | float | describe the morphology of the lesion |
| area | float | describe the morphology of the lesion |
| perimeter | float | describe the morphology of the lesion |
| area_perimeter_ratio | float | ratio of area and perimeter |
| shape_solidity | float | |
| laplacian_entropy | float | Laplacian energy features |
| sobel_entropy | float | Sobel energy features |
| glcm_contrast | float | capture occurrence of gray level pattern within the lesion |
| glcm_homogeneity | float | capture occurrence of gray level pattern within the lesion |
| histogram_egde | float | quantify edge sharpness along the lesion contour |
| mean_diff_in_out | float | |

Table 7: Heuristic Functions for Mammogram Tumor Classification

| Name | Heuristic Function | Description |
|---|---|---|
| hf_area | 1 if area $\geq 100000$ <br> $-1$ if area $\leq 30000$ <br> 0 otherwise | Large tumor area indicates malignant tumors |
| hf_diameter | 1 if diameter $\geq 400$ <br> $-1$ if diameter $\leq 200$ | High diameter indicates malignant tumors |
| hf_eccentricity | 1 if eccentricity $\leq 0.4$ <br> $-1$ if eccentricity $\geq 0.6$ <br> 0 otherwise | Low eccentricity indicates malignant tumors |
| hf_perimeter | 1 if perimeter $\geq 4000$ <br> $-1$ if perimeter $\leq 2500$ <br> 0 otherwise | High perimeter indicates malignant tumors |
| hf_max_intensity | 1 if max_intensity $\geq 70000$ <br> $-1$ if max_intensity $\leq 50000$ <br> 0 otherwise | High maximum intensity indicates malignant tumors |
| hf_mean_intensity | 1 if mean_intensity $\geq 45000$ <br> $-1$ if mean_intensity $\leq 30000$ <br> 0 otherwise | High mean intensity indicates malignant tumors |

Table 8: Heuristic Functions for Image Querying

| Name | Heuristic Function | Description |
|---|---|---|
| `hf_street` | 1 if `person` and `road`<br>−1 if `person` and `!road`<br>0 otherwise | Indicates if a street is present when person is present, doesn't assign a label when person is not present |
| `hf_vehicles` | 1 if `person` and `car`<br>−1 if `person` and `!car`<br>0 otherwise | Indicates if a car is present when person is present, doesn't assign a label when person is not present |
| `hf_positions` | −1 if `!bike` or `!person`<br>1 if $(\texttt{person.pos}-\texttt{bike.pos})\leq 1$<br>0 otherwise | Indicates if person and bike are close in the image |
| `hf_size` | 0 if `!bike` or `!person`<br>−1 if $(\texttt{person.area}-\texttt{bike.area})\geq 1000$<br>1 otherwise | Indicates if the difference in area of the bike and person is less than a threshold |
| `hf_number` | −1 if `!bike` or `!person`<br>1 if `num_persons = num_bikes`<br>−1 if `num_persons = num_bikes`<br>0 otherwise | Indicates if number of persons and bikes are equal |

Table 9: Heuristic Functions for Video Classification

| Name | Heuristic Function | Description |
|---|---|---|
| `hf_person_ball` | 1 if `person` and `ball`<br>−1 otherwise | Indicates if person and sports ball were present in any frame of the video |
| `hf_distance` | −1 if $\texttt{person.top}-\texttt{ball.bottom}\geq 2$<br>1 if $\texttt{person.top}-\texttt{ball.bottom}\leq 1$<br>0 otherwise | Indicates if the distance between person and sports ball is less than a threshold |
| `hf_ball_color` | 1 if `ball` and `ball.color −`<br>      $\texttt{basketball.color}\leq 80$<br>−1 otherwise | Indicates if the color of the ball is similar to the color of a basketball |
| `hf_temporal` | 1 if $\texttt{vertical\_distance}\geq 15$<br>−1 otherwise | Indicates if sufficient vertical distance was covered by the ball over frames |

Table 10: Heuristic Functions for Bone Tumor Classification

| Name | Heuristic Function |
|---|---|
| hf_daube | 1 if `histogram_164` $< 0.195545$ and `histogram_224` $< -0.469812$<br>$-1$ if `histogram_164` $< 0.195545$ and `histogram_201` $< 0.396779$<br>1 if `histogram_164` $< 0.195545$<br>$-1$ otherwise |
| hf_edge | $-1$ if `window_std` $< -0.0402606$<br>$-1$ if `window_median` $< -0.544591$<br>$-1$ if `scale_median` $< -0.512551$<br>1 otherwise |
| hf_lesion | $-1$ if `lesion_density` $< 0$<br>1 if `lesion_density` $> 1$ and `edge_sharpness` $< 0$<br>1 if `lesion_density` $> 5$ and `edge_sharpness` $> -1$<br>0 otherwise |
| hf_shape | $-1$ if `equiv_diameter` $< -0.3$<br>$-1$ if `equiv_diameter` $> 0$ and `area_perimeter_ratio` $< 0.5$ and `shape_solidity` $< 0.1$<br>1 if `equiv_diameter` $> 0$ and `area_perimeter_ratio` $< 0.5$ and `shape_solidity` $< 0.75$<br>1 if `equiv_diameter` $> 0$ and `area_perimeter_ratio` $> 1$<br>0 otherwise |
| hf_sobel_laplacian | $-1$ if `laplacian_entropy` $< 0.2$<br>1 if `laplacian_entropy` $> 0.4$ and `sobel_entropy` $< -0.75$<br>1 if `laplacian_entropy` $> 0.4$ and `sobel_entropy` $> -0$<br>0 otherwise |
| hf_glcm | $-1$ if `gclm_contrast` $< 0.15$ and `gclm_homogeneity` $< 0$<br>1 if `gclm_contrast` $< 0.15$ and `gclm_homogeneity` $> 0.5$<br>$-1$ if `gclm_contrast` $> 0.25$<br>0 otherwise |
| hf_first_order | $-1$ if `histogram_egde` $< 0.5$<br>1 if `histogram_egde` $> -0.3$ and `mean_diff_in_out` $< -0.75$<br>1 if `histogram_egde` $> -0.3$ and `mean_diff_in_out` $> -0.5$<br>0 otherwise |

We omit the descriptions of heuristic functions for bone tumor classification due to their complexity.