[Reviews · NeurIPS 2017]

Reviewer 1



The authors consider the setting where we wish to train a discriminative model using labels that generated using so-called heuristic functions, which in turn make use of primitive features. In order to generate a single label that combines multiple heuristics, the authors learn a probabilistic model (represented as a factor graph). This probabilistic model incorporates two types of structure. The method performs static analysis on the heuristic functions to identify cases where multiple functions make use of the same primitives. In order to capture correlations between primitives, the model learns pairwise similarities. The paper is very well written. The experiments which consider 4 datasets, demonstrate that knowing dependencies between primitive features does indeed significantly accuracy of the trained discriminative model in certain cases. ## Methodology In all the proposed methodology sounds reasonable to me. I am perhaps a little bit skeptical about the significance of the static analysis part of this work. Based on the fact that we are doing static analysis in the first place, it would appear that the authors assume a setting in which the user writes the heuristic functions, but makes use of black box primitives in doing so. Given that experiments consider settings in which there are < 10 primitives and < 10 heuristic functions, it seems to me that the user could easily specify interdependencies by hand, so walking the AST seems like a heavyweight solution for a relatively trivial problem. Given the above, I would appreciate a clearer articulation of the contributions made in this work (which is not in my area). If I understand correctly, learning binary dependencies between primitives is not novel here, so is the real contribution the static analysis part? If so, then I would like the authors to be a bit more explicit about what the limitations of their static analysis techniques are. If I understand correctly, the authors assume that all dependencies between heuristic functions arise in definitions of primitives that derive from other primitives and that each lambda accepts only one (possibly derived) primitive as input? ## Experiments The bone tumor result is a nice validation that generated labels increase classifier accuracy on the test set. Space allowing, could the authors perform the same experiment with other datasets (i.e. train the FS case on half of the data, and train the HF / HF+DSP models using the full dataset). ## Minor Comments - Figure 5: It would be helpful to explain in the main text HF means "only static analysis" whereas HF+DSP means "static analysis + pairwise similarities" - Figure 5: Could the authors indicate in the caption that the HF/HF+DSP are trained on additional data for which no labels are available. - Figure 4 is not called out in the text. - Bone Tumor: Could the authors explain a bit more clearly why knowing the number of primitives that each HF employs yields an improvement over the independent case? - when we looked only used -> when we only used

Reviewer 2



The authors propose using static analysis to improve the synthesis of generative models designed to combine multiple heuristic functions. They leverage the fact that expressing heuristics as programs often encodes information about their relationship to other heuristics. In particular, this occurs when multiple heuristics depend (directly or indirectly) on the same value, such as the perimeter of a bounding box in a vision application. The proposed system, Coral, infers and explicitly encodes these relationships in a factor graph model. The authors show empirically how Coral leads to improvements over other related techniques. I believe the idea is interesting and worth considering for acceptance. However, I have various queries, and points that might strengthen the paper: * I found it difficult at first to see how Coral relates to other work in this area. It might be useful to position the "Related Work" section earlier in the paper. It might also help to more explicitly describe how Coral relates to [2] (Bach et al.), which seems closest to this work -- in particular, there is quite a lot of similarity between the factor graphs considered by the two models. (Coral seems to be obtained by inserting the phi_i^HF and p_i nodes between the lambda_i and phi^DSP nodes, and by restricting only to pairwise terms in the phi^DSP factor.) * It isn't stated anywhere I could see that lambda_i \in {-1, 1}, but based on the definition of phi^Acc_i (line 147), this must be the case, right? If so, the "True" and "False" values in Figure 3 (a) should potentially be replaced with 1 and -1 for clarity. * Why does (i) (line 156) constitute a problem for the model? Surely over-specification is OK? (I can definitely see how under-specification in (ii) is an issue, on the other hand.) * What is the justification for using the indicator function I[p_i = p_j] for phi_ij^Sim? I would expect this hardly ever to have the value 1 for most primitives of interest. It is mentioned that, for instance, we might expect the area and perimeter of a tumour to be related, but surely we do not expect these values to be identical? (I note that similar correlation dependencies occur in [2] (Bach et al.), but these compare the heuristic functions rather than the primitives as here. Since the heuristic functions return an estimate of the label Y, it seems much more reasonable to expect their values might be equal.) * It is stated in line 176 that the supplementary materials contains a proof that "if any additional factor would have improved the accuracy of the model, then the provided primitive dependencies would have to be incorrect". However, I couldn't find this -- where exactly can it be found? Finally, some minor (possible) typos: * Line 133: "if an edge point" should be "if an edge points" * Figure 3 (a): The first "return False" has an extra indent * The double summation in the definition of P between lines 170 and 171 seems to double count certain terms, and is slightly different from the definition of phi^DSP between lines 169 and 170 * Line 227: "structure learning structure learning" * Line 269: "structure learning performed better than Coral when we looked only used static analysis to infer dependencies"

Reviewer 3



This paper builds on the work of Ratner et al. (NIPS 2016) and Bach et al. (ICML 2017) of using generative models and heuristics to cheaply obtain labeled data for problems where data is scarce. The contribution of this paper is use static analysis on these heuristic functions (HFs) to guide the structure learning. The insight is that many HFs will share primitives. These primitives are specified manually by the user and are called Domain-specific primitives (DSP). Modeling the dependencies between these HFs in the generative model leads to an improvement on F1 score for several datasets when the labels from that generative model are later used as training data in a discriminative model. The approach of using static analysis for this problem is a novel contribution. Static analysis here means the Heuristic Functions (HF) meaning the source code was analyzed to figure out which primitives they share. The work is complete with experimental results which support the claims of the paper. The experiments seem to use a small number of DSPs and HFs. This makes the experiments less than ideally support what we getting from use static analysis to extract the higher order dependencies. The paper suggests pairwise dependencies between HFs are modeled as a hedge if the static analysis fails. What is nice is that in all the datasets used the DSP factor always seemed to make no difference to the F1-score. It would have been nice to see a dataset where that factor really makes a difference. The experimental results are also interesting in that for the Bone Tumor dataset the heuristics outperform on F1-score the fully-supervised case. I'd be curious if there are other datasets that exhibit this behavior or is this just an artifact of the Bone Tumor dataset being smaller. The paper is clearly written, though it does build significantly on recent work. That makes the paper challenging to understand by itself. Minor details: Is F1 the best metric to use here? Also why are some of the values in Table 1 in terms of F1 score and others use accuracy? DSP is really overloaded for me as Digital Signals Processing. Is there any other abbreviation that could have been chosen? The Primitive Set looks like a record type. Are there any restrictions on what constitutes a Primitive Set?